# Machine Learning Techniques for Developing Remotely Monitored Central Nervous System Biomarkers Using Wearable Sensors: A Narrative Literature Review

**DOI:** 10.3390/s23115243

**Published:** 2023-05-31

**Authors:** Ahnjili ZhuParris, Annika A. de Goede, Iris E. Yocarini, Wessel Kraaij, Geert Jan Groeneveld, Robert Jan Doll

**Affiliations:** 1Centre for Human Drug Research (CHDR), Zernikedreef 8, 2333 CL Leiden, The Netherlands; azhuparris@chdr.nl (A.Z.);; 2Leiden Institute of Advanced Computer Science (LIACS), Snellius Gebouw, Niels Bohrweg 1, 2333 CA Leiden, The Netherlands; 3Leiden University Medical Center (LUMC), Albinusdreef 2, 2333 ZA Leiden, The Netherlands; 4The Netherlands Organisation for Applied Scientific Research (TNO), Anna van Buerenplein 1, 2595 DA, Den Haag, The Netherlands

**Keywords:** machine learning, biomarker, wearables, smartphones, mHealth, remote monitoring, central nervous system, clinical trials

## Abstract

Background: Central nervous system (CNS) disorders benefit from ongoing monitoring to assess disease progression and treatment efficacy. Mobile health (mHealth) technologies offer a means for the remote and continuous symptom monitoring of patients. Machine Learning (ML) techniques can process and engineer mHealth data into a precise and multidimensional biomarker of disease activity. Objective: This narrative literature review aims to provide an overview of the current landscape of biomarker development using mHealth technologies and ML. Additionally, it proposes recommendations to ensure the accuracy, reliability, and interpretability of these biomarkers. Methods: This review extracted relevant publications from databases such as PubMed, IEEE, and CTTI. The ML methods employed across the selected publications were then extracted, aggregated, and reviewed. Results: This review synthesized and presented the diverse approaches of 66 publications that address creating mHealth-based biomarkers using ML. The reviewed publications provide a foundation for effective biomarker development and offer recommendations for creating representative, reproducible, and interpretable biomarkers for future clinical trials. Conclusion: mHealth-based and ML-derived biomarkers have great potential for the remote monitoring of CNS disorders. However, further research and standardization of study designs are needed to advance this field. With continued innovation, mHealth-based biomarkers hold promise for improving the monitoring of CNS disorders.

## 1. Introduction

### 1.1. Motivation

Disorders that are affected by the Central Nervous System (CNS), such as Parkinson’s Disease (PD) and Alzheimer’s Disease (AD), have a significant impact on the quality of life of patients. These disorders are often progressive and chronic, making long-term monitoring essential for assessing disease progression and treatment effects. However, the current methods for monitoring disease activity are often limited by accessibility, cost, and patient compliance [1,2]. Limited accessibility to clinics or disease monitoring devices may hinder the regular and consistent monitoring of a patient’s condition, especially for patients living in remote areas or for those who have mobility limitations. Clinical trials incur costs related to personnel, infrastructure, and equipment. A qualified healthcare team, including clinical raters, physicians, and nurses, contributes to personnel costs through salaries, training, and administrative support. Trials involving specialized equipment for measuring biomarkers can significantly impact the budget due to costs associated with procurement, maintenance, calibration, and upgrades. Furthermore, infrastructure costs may increase as suitable facilities are required for data collection during patient visits and equipment storage. Patient compliance poses challenges for disease monitoring, as some methods require patients to adhere to strict protocols, collect data at specific time intervals, or perform certain tasks that can be challenging for patients to execute. Low or no compliance can lead to incomplete or unreliable monitoring results, which in turn can hinder the reliability of the assessments. Given these limitations, there is a growing interest in exploring alternative approaches to monitoring CNS disorders that can overcome these challenges. The increasing adoption of smartphones and wearables among patients and researchers offers a promising avenue for remote monitoring.

Patient-generated data from smartphones, wearables, and other remote monitoring devices can potentially complement or supplement clinical visits by providing data during evidence gaps between visits. As the promise of mobile Health (mHealth) technologies is to provide more sensitive, ecologically valid, and frequent measures of disease activity, the data collected may enable the development and validation of novel biomarkers. The development of novel ‘digital biomarkers’ using data collected from electronic Health (eHealth) and mHealth device sensors (such as accelerometers, GPS, and microphones) offers a scalable opportunity for the continuous collection of data regarding behavioral and physiological activity under free-living conditions. Previous clinical studies have demonstrated the benefits of smartphone and wearable sensors to monitor and estimate symptom severity associated with a wide range of diseases and disorders, including cardiovascular diseases [3], mood disorders [4], and neurodegenerative disorders [5,6]. These sensors can capture a range of physiological and behavioral data, including movement, heart rate, sleep, and cognitive function, providing a wealth of information that can be used to develop biomarkers for CNS disorders in particular. These longitudinal and unobtrusive measurements are highly valuable for clinical research, providing a scalable opportunity for measuring behavioral and physiological activity in real-time. However, these approaches may carry potential pitfalls as the data sourced from these devices can be large, complex, and highly variable in terms of availability, quality, and synchronicity, which can therefore complicate analysis and interpretation [7,8]. Machine Learning (ML) may provide a solution to processing heterogenous and large datasets, identifying meaningful patterns within the datasets, and predicting complex clinical outcomes from the data. However, the complexities involved in developing biomarkers using these new technologies need to be addressed. While these tools can aid the discovery of novel and important digital biomarkers, the lack of standardization, validation, and transparency of the ML pipelines used can pose challenges for clinical, scientific, and regulatory committees. 

### 1.2. What Is Machine Learning

In clinical research, one of the primary objectives is to understand the relationship between a set of observable variables (features) and one or more outcomes. Building a statistical model that captures the relationship between these variables and the corresponding outputs facilitates the attainment of this understanding [9]. Once this model is built, it can be used to predict the value of an output based on the features. 

ML is a powerful tool for clinical research as it can be used to build statistical models. A ML model consists of a set of tunable parameters and a ML algorithm that enables the generation of outputs based on given inputs and selected parameters. Although ML algorithms are fundamentally statistical learning algorithms, ML and traditional statistical learning algorithms can differ in their objectives. Traditional statistical learning aims to create a statistical model that represents causal inference from a sample, while ML aims to build generalizable predictive models that can be used to make accurate predictions on previously unseen data [10,11]. However, it is essential to recognize that while ML models can identify relationships between variables and outcomes, they may not necessarily identify a causal link between them. This is because even though these models may achieve good performances, it is crucial to ensure that their predictions are based on relevant features rather than spurious correlations. This enables the researchers to gain meaningful insights from ML models while also being aware of their inherent limitations.

While ML is not a substitute for the clinical evaluation of patients, it can provide valuable insights into a patient’s clinical profile. ML can help to identify relevant features that clinicians may not have considered, leading to better diagnosis, treatment, and patient outcomes. Additionally, ML can help to avoid common pitfalls observed in clinical decision making by removing bias, reducing human error, and improving the accuracy of predictions [12,13,14,15]. As the volume of data generated for clinical trials and outside clinical settings continues to grow, ML’s support in processing data and informing the decision-making process becomes necessary. ML can help to uncover insights from large and complex datasets that would be difficult or impossible to identify manually.

To develop an effective ML model, it is necessary to follow a rigorous and standardized procedure. This is where ML pipelines come in. Table 1 showcases an exemplary ML pipeline, which serves as a systematic framework for automating and standardizing the model generation process. The pipeline encompasses multiple stages, as defined by the authors, to ensure an organized and efficient approach to model development. First, defining the study objective guides the subsequent stages and ensures the final model meets the desired goals. Second, raw data must be preprocessed to remove errors, inconsistencies, missing data, or outliers. Third, feature extraction and selection identifies quantifiable characteristics of the data relevant to the study objective and extracts them for use in the ML model. Fourth, ML algorithms are applied to learn patterns and relationships between features, with optimal configurations identified through iterative processes until desired performance metrics are achieved. Finally, the model is validated against a new dataset that is not used in training to ensure generalizability. Effective reporting and assessment of ML procedures must be established to ensure transparency, reliability, and reproducibility.

### 1.3. Objectives

The objective of this narrative literature review is to provide an overview of the ML practices used in studies that use mHealth technologies and ML to develop novel biomarkers for clinical trials. In this review, each component of the ML pipeline has a dedicated section. Based on the results obtained from the review process, each ML component section provides a comprehensive analysis and discussion of the most common and notable practices. These sections delve into the motivations behind these practices, their limitations, and their overall impact on the ML pipeline. This review will not provide precise recommendations for best practices, as much of the research in this area is new and quickly evolving. Rather, the recommendation section discusses the approaches for standardization and validation procedures that are necessary for the development of ML biomarkers to ensure the effectiveness and acceptance of these biomarkers by clinical, scientific, and regulatory committees. 

## 2. Methods

### 2.1. Information Sources and Search Strategy

Given the wide range of study designs and clinical populations that use smartphones and wearables to collect data, we used the Joanna Briggs Institute (JBI) guidelines to develop a search strategy [16]. Based on an initial limited search of online databases for clinical trials that report using mHealth devices and ML, we developed a custom keyword strategy and performed an in-depth search in PubMed, IEEE Xplore, and CTTI (Table 2). The search terms for the CNS disorder terms were based on the National Library of Medicine’s CNS MeSH descriptor data [17]. The relevant papers were selected based on the title and abstract. Finally, other literature review studies that explore the same questions were reviewed; the references cited by these studies were then identified and reviewed if they met our criteria. The date range for the search was between 1 January 2012 and 31 December 2022. The search was conducted on 7 January 2023. 

### 2.2. Inclusion Criteria

The authors adopted the Population, Intervention, Comparator, Outcomes, Study type (PICOS) framework to define the inclusion and exclusion criteria [13] (Table 3). The studies included were restricted to those involving participants diagnosed with CNS disorders who were remotely monitored under free-living conditions. The intervention and device criteria were limited to passive data collected from smartphones and other non-invasive remote monitoring sensors, whereas data collected using active engagement from participants, such as disposable blood tests or small scales, were excluded. As we chose to focus on ML pipelines, we selected studies in which a statistical model was used to analyze a dataset and could potentially be used to generate future predictions using an independent dataset. Therefore, traditional statistical models such as linear or logistic regression were included, but statistical models such as ANOVA and correlation analyses were not included. Further, as the focus was on the development and validation of ML models, we did not include studies that did not report on model performance. 

### 2.3. Data Extraction

Two authors conducted the data extraction following the inclusion criteria, and the results were reviewed by the remaining authors. Data relating to the database source, title, DOI, publication year, trial setting or scenario, objective, devices used, data collection period, number of participants, inclusion of healthy controls, data processing steps, feature engineering, feature selection, machine learning models used, hyperparameters and hyperparameter optimization, model performance, and validation procedure were extracted. The comprehensive data extraction and review conducted by the authors encompassed various essential aspects of the studies, ensuring a thorough analysis of the database source, trial details, data processing steps, machine learning models, and validation procedures.

## 3. Results

### 3.1. Study Selection

Our initial keyword search revealed a total of 2310 articles that utilized digital phenotyping devices, such as smartphones and wearables, in a clinical study and applied ML techniques. After screening the titles and abstracts based on our predefined criteria, we narrowed down the articles to 66 studies, which were used for our analysis. Figure 1 provides an overview of the complete selection process.

### 3.2. Study Characteristics

For each of the 66 studies, we extracted information about the clinical population and the ML pipeline that was used to develop the digital biomarkers. We found that only half of the studies included healthy controls (N = 34). As seen in Figure 2, Parkinson’s disease (PD) (N = 27) was the most prevalent disorder identified in our search, followed by Bipolar Disorder (BD) (N = 11), and Unipolar Depression or Major Depressive Disorder (MDD) (N = 9). The sample size of the selected studies was heterogenous, ranging from 7 to 6221 participants (Figure 3). Overall, our review provides a comprehensive overview of the characteristics of studies that have utilized mHealth devices and ML techniques, which can help inform future research in this field. In the following sections, we addressed how the selected studies approached the construction of their ML pipelines.

## 4. Missing and Outlier Data

Missing and outlier data are commonly encountered problems for remote sensing clinical trials. Missing data can be the result of device charging frequency, device robustness, and participant compliance [18]. Outliers can be the result of sensor or device dysfunction or malfunction, incorrect data entry, and incorrect classifications [19]. Data preprocessing, which refers to the dropping or manipulation of data, is required for identifying and removing redundant or irrelevant data and for cleaning the data prior to analysis. Without preprocessing, learning from an imperfect dataset can influence the prediction accuracy of the models [20]. In this section, we address how the selected studies preprocessed their raw data by treating their missing data and outliers, and the limitations of doing so. 

### 4.1. Handling of Missing Data

Missing data can be Missing Completely at Random (MCAR), Missing at Random (MAR), and Missing Not at Random (MNAR) [21]. MCAR assumes that each observation has the same probability of being included or being missed; therefore, there is no difference in the characteristics between participants or observations without missing data and those with missing data. For example, data may be missed due to the battery of the smartphone running out. MAR assumes that missing data may have systematic differences between the missing and non-missing data; however, the cause of the missing data can be explained by the non-missing data. For example, a smartphone may have more missing values when the smartphone battery is low. If the battery percentage is known during the data acquisition, researchers can verify the probability of acquiring missing data depending on the battery percentage. MNAR assumes that missing data are caused by unknown reasons. For example, smartphone sensors may be gradually worn down, which therefore creates more missing data over time. The type of missing data present in the dataset influences whether a researcher should ignore, exclude, or impute the missing data. 

Among the selected studies, we found that only 21 of the studies reported the quantity of missing data acquired. Only 29 studies reported how they handled their missing data. We found that complete-case analysis and imputation were the most popular. We identified 14 studies that report using complete-case analysis [22,23,24,25,26,27,28,29,30,31,32,33,34,35,36]. Complete-case analysis (otherwise known as listwise deletion) is the deletion of an observation involving one or multiple elements of missing data [26,37,38]. While complete-case analysis is the simplest approach to handle missing data, it does reduce the sample size and statistical power of the analysis [39] and can potentially lead to bias if the data are not MCAR [40]. Imputation is the statistical process of replacing missing data with substituted inferred values [41]. We identified studies that imputed their missing data using linear interpolation [29,42,43], forward filling [44], −1 [45], zeros, median, means, and the most frequent value in the column [24,46]. The advantage of imputation is that it enables researchers to use all observations in the dataset. However, the inclusion of imputed values can lead to a false impression of the number of complete cases and reduce the variance in the dataset [47,48,49]. 

### 4.2. Identification of Outliers

Aggarwal’s *Data mining: the textbook* states that it is the subjective definition of the researcher that defines an outlier [50]. In cases where the outlier data were discussed in the selected studies, we found that researchers customized their definition of outliers by either defining a range of acceptable values [32] or by defining a threshold based on the mean and standard deviation [51,52,53]. Visual inspection by the researchers or the optimization of different threshold mechanisms can both be used to define the boundaries of normal or outlier data [54,55]. Maleki et al. defined outliers as observations that were most likely the result of measurement errors [36]. In terms of the handling of outliers, we only identified six studies that explicitly stated that outliers were excluded [26,30,51,52,53,56]. 

## 5. Feature Engineering

### 5.1. Feature Scaling

Feature scaling is used to normalize the ranges of the features in a dataset [57]. Several feature engineering techniques and ML models (such as Principal Component Analysis and Linear Regression) calculate the distances between two observations. If one feature has a broader range of values compared to the other features, the calculated distances will be heavily influenced by this feature [58]. Therefore, the ranges of all the features should be normalized or standardized so that each feature is appropriately and proportionally considered with respect to the estimated distances [57]. Feature normalization is a common scaling method for rescaling the features into a bounding range using the minimum and maximum values, for example, between 0 and 1. Normalization is an ideal approach when the distribution of the data is not Gaussian, as normalization preserves the original distribution of the data. However, normalization uses minimum and maximum values to define ranges. This makes the method sensitive to outliers [57,59]. Alternatively, feature standardization, also known as z-score normalization, is a method for rescaling the data to fit a standard normalized distribution by using the mean and standard deviation and does not define a bounding range. Consequently, the standardization approach is not sensitive to outliers as it has no bounding range [57,59]. Normalization, log-transformation, and standardization have been reported in a small selection of the selected studies [26,27,36,60,61]. 

### 5.2. Expert Feature Engineering

Feature engineering is the process of constructing (new) features from the raw data or existing features while maintaining the original patterns and information in the data [62]. The newly engineered features can be added to or replace features in the original dataset. Engineering of the features can speed up the model performance, improve learning accuracy, and ease the interpretability of the model. The latter is particularly important for clinical trials [63]. Features can be engineered manually by relying on domain-knowledge or automatically by using statistical models, such as Principal Component Analysis (PCA) and Deep Learning (DL) [62,63,64]. All features aim to increase the separability between the classes or signals, which in turn reduces noise in the dataset. While expert engineered features are easy to interpret and explain and have been widely used in the development of digital biomarkers, these features are typically task- or population-dependent. Due to intra-class variability, some clinically relevant characteristics may be exhibited differently by different individuals (such as different symptom profiles among patients with the same diagnosis). Furthermore, expert engineered features may not be sufficient for representing the most important characteristics of complex patterns and can be time-consuming to acquire, especially when handling large-scale datasets [65,66]. As clinical data has expanded in terms of diversity, availability, and complexity, the aforementioned techniques may be insufficient for developing generic features. In the following sections, we address the notable and generic procedures used to perform feature engineering.

### 5.3. Signal Processing

To monitor changes in the physical activity of study participants using time series data collected from wearable sensors, signal processing is necessary to detect, clean, and analyze the components of interest. The feature extraction technique used is influenced by the sensor type, study objectives, and signal quality. Typically, signal features are extracted from the frequency, time, or cepstrum domain [67]. Frequency domain features show the prominence of a signal within a given frequency, whereas time-domain features show the changes in the signal of time. Cepstrum domain features represent the rate of change in the different frequency bands. The analysis of the frequency, time, or cepstrum domain features is not mutually exclusive. We identified studies that use both time- and frequency-based features for the estimation of gait speed [68], speech-tasks [69], seizure detection [70], tremor detection [71], and FOG detection [72]. In particular, Tougui et al. built 138 voice-related features extracted from the cepstral, frequency, and time domains [24]. In sum, time series data collected from wearable sensors can be used to monitor the physical activity of study participants, but signal processing is necessary to extract meaningful features. Different feature extraction techniques can be used depending on the sensor type, signal quality, and study objectives. The analysis of these features is not mutually exclusive, and studies that use multiple domains for different clinical applications have been identified. 

### 5.4. Principal Component Analysis

A common linear dimensionality reduction technique for feature engineering and selection is Principal Component Analysis (PCA) [28,73,74]. PCA is used to sufficiently explain a high-dimensional dataset through a few principal components and, therefore, to reduce a high-dimensional dataset to one of fewer dimensions [75]. To this purpose, PCA converts a set of correlated features into a set of uncorrelated features by utilizing orthogonal transformation [75]. The principal components enable a reduction in the feature space by creating a linear combination of the original features, which consequently reduces the storage space and reduces the learning time. Therefore, the periodic components within a concurrent time series dataset can be isolated using PCA, which can subsequently be used to identify any underlying patterns within the dataset. It is important to note that PCA assumes that the data are normally distributed and is sensitive to feature variance [75,76]. Consequently, features with larger ranges will dominate features with smaller ranges. To make the variables comparable, transformation of the data prior to PCA is required [75,76]. Of the studies selected, PCA was used to engineer and select features from times series data sourced from waist-worn triaxial accelerometers and wearable activity trackers [28,73,74]. However, the limitations of PCA are its sensitivity to missing data and outliers and the limited interpretation of the original features. Hence, this observation highlights the need for thorough data preprocessing prior to using PCA. 

### 5.5. Clustering

A clustering algorithm is a common feature engineering method that assigns similar observations to a single cluster and assigns dissimilar observations to another [77]. While PCA compresses the features into principal components, clustering compresses the individual observations into clusters. The grouping of similar observations can improve the model’s ability to discriminate between classes [78]. Clustering algorithms, more specifically DBSCAN and K-means clustering, have been deployed in smartphone GPS systems and Wi-Fi-network sensors to extract meaningful location features such as frequented location clusters [79], location patterns [80], and mobility patterns [81]. These studies demonstrate that clustering algorithms are a powerful method for reducing the number of observations into a smaller number of artificial variables that account for the variance within the dataset. 

### 5.6. Deep Learning

The performance of ML models can be limited by the development of manual and arbitrary features, and this potential obstacle can be overcome by DL algorithms. DL algorithms eliminate the need for manual feature engineering, as the DL layers can translate the data into more compact and intermediate abstractions of the data, which in turn can be used as features to predict the final output [82]. While DL can reduce the need for manual data preprocessing and feature extraction, which can potentially improve the generalizability and robustness of a model, the interpretation of the DL model is difficult, as the abstracted features may not be explainable by clinicians. However, it is important to note that the discriminative power of the DL-derived abstractions is strongly influenced by the architecture of the DL algorithm, which is also dependent on the trial-and-error process [59]. Due to DL’s representation learning, DL is data-hungry, and therefore requires more data than other ML algorithms [83,84]. For clinical trial data, because of technological limitations and small sample sizes, there may not be enough data to train a sufficiently representative DL model [76,83]. 

Four studies used DL to engineer features using time series data [23,85,86,87]. These models were used to extract gait features from accelerometer data [85,87] and tremor characteristics from IMU data [23,86]. However, it should be noted that the DL models do not always outperform the ‘shallow learning’ models, as shown in a study by Juen et al., in which smartphone accelerometers were used to predict natural walking speed and distance during a six-minute walk test [85]. 

## 6. Feature Selection

In recent decades, high-dimensional clinical datasets have relied on feature selection [88]. Feature selection is the process of selecting a subset of the most informative features that will be processed by the ML algorithm [89]. Reducing the features for analysis has both computational and practical benefits. Selecting features can limit storage requirements, increase the algorithm processing speed, increase the interpretability of a model, and improve model performance. 

### 6.1. Overfitting and Underfitting

Overfitting and underfitting are common pitfalls for ML models. Overfitting refers to when a ML model fits too well to its training dataset and is unable to generalize its patterns to unseen data. This problem can occur when the training dataset is small and not representative of the overall potential data distribution. Additionally, if the training dataset contains many outliers, the ML model may also fit the outlier data. Underfitting occurs when the trained ML model is too simple; therefore, it cannot identify the relationship between the features and the outputs. Underfitted models will perform poorly for both the train and validation datasets. To address overfitting, reducing the number of features considered by the model or updating the model architecture to include fewer features can be effective [90]. Underfitting can be improved by adding more features considered by the model or by updating the model architecture to increase the complexity of the feature space [90].

Feature selection identifies the most important features in the dataset and eliminates the irrelevant ones, which thereby reduces noise. However, it is important to strike a balance, as strict feature selection may remove important signals from the data. Therefore, selecting the optimal set of features is important for preventing over- and underfitting. In the following sections, we will elaborate on the three general methods of feature selection that are suitable for ML models [75]. 

### 6.2. Filter Methods

Filter methods are used during preprocessing prior to training the ML model. Filtering involves removing features based on domain knowledge, missing data, low variance, or correlation [89,91,92]. As filter methods are independent of any model that is to be used in later steps, they are typically faster to implement and reduce the need for repeating feature selection for different ML models. In our selected studies, we found five studies that used Analysis of Variance (ANOVA), Pearson’s Correlation, or Spearman’s Correlation to identify features that were statistically significant predictors of the outcomes [24,93,94,95,96]. *p*-value based feature selection, while commonly used in clinical studies, is not always suitable for training a ML model. The use of *p*-values to identify statistically significant features was a popular approach that relied on the belief that insignificant features were not informative. However, important features can be missed when sample sizes are small. Furthermore, *p*-values can be biased towards low values due to the increased risk of type 1 errors during multiple comparisons, which in turn increases the probability of random variables being included into the final statistical model [97,98]. Additionally, p-value based feature selection methods may be based on certain assumptions that may not be applicable to ML models, such as assuming that the distribution of scores for the groups among the independent variables are the same [99].

We wanted to highlight one filtering method identified in our selected studies: Relief [100]. Relief is a feature selection technique that also ranks features and selects only the top-scoring features; however, it is notably sensitive to feature interactions [101,102]. Yaman et al., first obtained 177 speech-related features and used Relief to select 66 most predictive vocal biomarkers for the classification of PD [103]. Rodriguez-Molinero used Relief to select frequency features that were subsequently used to predict gait disturbances among PD patients [104]. Overall, Relief has demonstrated its effectiveness in selecting relevant features in various studies related to the prediction of PD using high-dimensional clinical datasets.

### 6.3. Embedded Methods

The embedded method is a feature selection technique integrated into the ML algorithm itself and is commonly seen in penalized regression [105]. Penalized regression algorithms aim to learn the optimal coefficients for each feature by minimizing its loss function. Regularization (also known as penalization) limits the learning process of the model by increasing the penalty of the loss function [106]. The two common penalized regression methods, identified in the selected studies, are LASSO (also known as L1 penalization) (N = 9) [22,24,29,33,42,95,100,101,107,108] and Ridge (L2 penalization) (N = 2) [109,110]. An advantage of LASSO is that it eliminates non-informative features by reducing their coefficients to zero. The first limitation of LASSO is that, if the number of features *f* is greater than the number of observations o, LASSO will select a maximum of o predictors as non-zeros, regardless of the relevance of other features. The second limitation is that LASSO also suffers from collinearity; hence, if two or more variables are highly correlated, then LASSO will randomly select one feature and penalize the other correlated features. A disadvantage of Ridge is that it only reduces the weights of the non-informative features by reducing their coefficients towards zero, but it never reduces the number of variables. Therefore, all predictors are included in the final model. However, because of this approach, Ridge protects ML models from overfitting [111]. 

### 6.4. Wrapper Methods

Wrapper methods rely on a stand-alone model to select features, but the performance of the selected features is reflected in the performance of the trained model [112]. The wrapper method algorithms tend to be greedy search algorithms that aim to select the optimal feature subset by iteratively selecting the features based on ML performance. As the wrapper method is an iterative process and the model must be evaluated on each feature subset combination, this method is computationally expensive. Wrapper-based feature selection can be completed by ranking the features in terms of relative importance using a ML model (such as decision trees or random forests) [88,101,113]. We identified a handful of feature ranking methods that include two stepwise regression techniques: Forward Selection and Backwards Elimination [29,36,52,114,115,116], as well as Recursive Feature Selection (RFE) [30,117]. Forward selection starts the modelling process with zero features and adds a new feature to the model incrementally, each time testing for statistical significance. Backwards elimination starts the modelling process with all features and incrementally removes each feature to evaluate its relative importance in predicting the model output [97,118]. RFE fits a model, ranks the features, and removes the least informative features and continues to remove features until a predefined number of features is met [64,119,120]. Senturk et al., illustrated that RFE-based feature selection increased the prediction accuracy of ANN, CART, and SVM when using vocal data to classify a PD diagnosis [121].

## 7. Machine Learning Algorithms

ML algorithms build a statistical model based on a training dataset, which can subsequently be used to make predictions about a new, unseen dataset. ML algorithms have been used in a wide variety of clinical trial applications, such as the classification of a diagnoses, classification of physical or mental state (such as a seizure or mood), and the estimation of symptom severity. Within the realm of clinical research, ML algorithms can be broadly divided into two learning paradigms: supervised and unsupervised learning [122]. In this section, we will discuss the model objectives of supervised and unsupervised learning and the specific ML models used to achieve these model objectives. 

Supervised ML algorithms use labeled data to map the patterns within a dataset to a known label, while unsupervised ML algorithms do not [123]. Rather, the unsupervised ML algorithms learn the structure present within a dataset without relying on annotations. Supervised learning can be used to automate the labelling process, detect disease cases, or predict clinical outcomes (such as treatment outcomes). There are scenarios when experts or participants can provide labelled data; however, it can become labor-intensive or time-consuming to label every observation. For example, a supervised learning algorithm trained to classify human sounds can be used to automatically annotate and quantify hours of coughs [124] and instances of crying [125]. These algorithms can also be used to differentiate between clinical populations and control participants [95] to identify known clinical population subtypes [23] or classify a clinical event (such as a seizure or tremor) [126]. The majority of our selected studies (N = 38) used a clinician to provide the label data. Some studies (N = 22) used a combination of a clinician and self-reported label data, and six studies solely relied on self-reported assessments. Unsupervised ML algorithms can be used to investigate the similarities and differences within a dataset without human intervention. This makes it the ideal solution for exploratory data analysis, subgroup phenotype identification, and anomaly detection. Among digital phenotyping studies, unsupervised learning has been used to identify location patterns [81] and classify sleep disturbance subtypes using wrist-worn accelerometer data [127].

It is important to recognize that unsupervised and supervised methods are not mutually exclusive, and they can be effectively combined. For instance, unsupervised methods can be employed to extract a meaningful latent representation of the input data. Subsequently, these latent vectors, along with the original inputs, can be used as inputs for a supervised model. This type of approach is commonly observed when applying techniques such as PCA, clustering, or other dimensionality reduction methods [29,73,74,128]. By combining unsupervised and supervised methods, valuable information can be extracted from the data and used to enhance the performance and interpretability of the overall model.

In clinical research, supervised ML algorithms have been used to classify class labels or estimate scores. Classification algorithms learn to map a new observation to a pre-defined class label. These algorithms can be used to classify patient populations and patient population subtypes and identify clinical events. Regression algorithms learn to map an observation to a continuous output. These algorithms are commonly used to estimate symptom severity [129], quantify physical activity, and forecast future events [130]. Among the selected papers that were focused on the classification of a diagnosis or state, the four most common algorithms were Random Forest, Support Vector Machine, Logistic Regression, and k-Nearest Neighbors (Figure 4). Some additional classification algorithm families identified were Naïve Bayes, Ensemble-based methods (including Decision Trees, Bagging, and Gradient Boosting), and Neural Networks (such as Convolutional, Artificial, and Recurring Neural Networks). The three most common algorithms for the regression-focused papers were Linear Regression (including linear mixed effects models), Support Vector Machine, and k-Nearest Neighbors (Figure 4). We found that most studies only considered or reported a single ML algorithm (N = 32). Additionally, 29 of the studies considered or reported two to five ML algorithms, and the remaining 5 studies considered six or more. The following section provides an overview of the most widely used machine learning models, their properties, advantages, and disadvantages. In addition, we discuss some notable off-the-shelf ML approaches and some custom-built ML methods such as transfer learning, multi-task learning, and generalized and personalized models. 

### 7.1. Tree-Based Models

A Decision Tree (DT) is a supervised non-parametric algorithm that is used for both classification and regression. A DT algorithm has a hierarchical structure in which each node represents a test of a feature, each branch represents the result of that test, and each leaf represents the class label or class distribution [131,132]. A Random Forest (RF) algorithm is a supervised ensemble learning algorithm consisting of multiple DTs that aims to predict a class or value [133]. Ensemble learning algorithms use multiple ML algorithms to obtain a prediction [134]. Tree-based models have several benefits. As each tree is only based on a subset of features and data and because they make no assumptions about the relationship between the features and distribution, they are not sensitive to collinearity between features, can ignore missing data, and are less susceptible to overfitting (for multiple trees), making the model more generalizable [135]. Another advantage of RF and DT models is that they can support linear and nonlinear relationships between the dependent and independent variables [136]. Further, as the design of the RF models can be interpreted in terms of feature importance and proximity plots, the interpretability of the RF model is feasible. However, a limitation of using tree-based models is that small changes in the data can lead to drastically different models. Additionally, the more complicated a tree-based model becomes, the less explainable a model becomes. However, pruning the trees can help to reduce the complexity of the model.

According to the selected studies, RF is a versatile and powerful model used for classification and regression tasks across multiple datatypes and populations. RF models have been used for the classification of diagnoses among PD patients [107,110], Multiple Sclerosis [34,118], and BD and unipolar depressed patients [45,61]. It is also a popular classification model for the classification of states or episodes, such as the detection of flares among Rheumatoid Arthritis or Axial Spondylarthritis patients [32] and tremor detection among PD patients [137], to quantify physical activity among cerebral palsy patients [138] and detect the moods of BD patients [69,139]. RF regression algorithms have also been used to predict anxiety deterioration among patients who suffer with anxiety [140]. 

### 7.2. Support Vector Machines 

A Support Vector Machine (SVM) is a supervised algorithm that is used for classification and regression tasks. The objective of a SVM is to identify the optimal hyperplane based on the individual observations, also known as the support vectors. For SVM regression, the optimal hyperplane represents the minimal distance between the hyperplane and the support vectors. Whereas for SVM classification, the objective is to find the hyperplane that represents the maximum distance between two classes [141]. The hyperplanes can separate the classes in either a linear or non-linear fashion [136]. Given that SVMs are influenced by the support vectors closest to the hyperplanes, SVMs are less influenced by outliers, making them more suitable for extreme case binary classification. The performance of a SVM can be relatively poor when the classes are overlapping or do not have clear decision boundaries. This makes SVMs less appealing for classification tasks as inter class similarity is low. SVMs are computationally demanding models as they compute the distance between each support vector; hence, SVMs do not scale well for large datasets [142]. 

SVM classifiers have been used to classify clinical populations (e.g., facial nerve palsy and their control participants) [143]. SVM classifiers have also been used to classify events or states, such as detecting gait among PD patients [104] and classifying seizures among epileptic children [144]. We identified studies that used SVM regression to estimate motor fluctuations and gait speed among PD and Multiple Sclerosis patients, respectively [74,145]. 

### 7.3. k-Nearest Neighbors 

A k-Nearest Neighbor (k-NN) algorithm is a non-parametric supervised learning approach that can be used for multi-class classification and regression tasks. Classification k-NN algorithms determine class membership by the plurality vote of its nearest neighbors. They can estimate the continuous value of an output by calculating the average value of its nearest neighbors [136]. Given this, the quality of predictions is not only dependent on the amount of data but also on the density of the data (the number of points per unit). K-NN is simple to implement, intuitive to understand, and robust to noisy training data. However, the disadvantage is that k-NN is computationally slow when it is faced with large multi-dimensional datasets. Further, k-NN does not work well with imbalanced datasets, as under- or over-represented datapoints will influence the classification [146]. 

The most popular application for k-NN algorithms is for wearable-based time series data. K-NN classification models have been used to classify PD and healthy controls [24], classify tremor severity [147], predict acute exacerbations of chronic obstructive pulmonary disease (AECOPD) [44], and identify mood stability among BD and MDD patients [33,69,148]. Using wearable data, k-NN regression models have been used to predict the deterioration of symptoms associated with anxiety disorder [140]. 

### 7.4. Naïve Bayes 

A Naïve Bayes (NB) classifier is a supervised multi-class classification algorithm. NB classifiers calculate the class conditional probability—the probability that a datapoint belongs to a given class in the data [141,149]. NB classifiers are computational efficient algorithms; thus, they are suitable for real-time predictions, scale well for larger datasets, and can handle missing values. A limitation of NB is that it assumes that all features are conditionally independent; hence, it is recommended that collinear features are removed in advance. Another limitation is that when new feature-observation pairs do not resemble the data in the training data, the NB assigns a probability of zero to that observation. This approach is particularly harsh, especially when dealing with a smaller dataset. Hence, the training data should represent the entire population.

As NB classifiers help form classification models, we found that NB classifiers have been used for the classification of tremors or for freezing gait among PD patients [52], as well as to classify flares among Rheumatoid Arthritis and Axial Spondylarthritis patients [32] and classify bipolar episodes and mood stability among BD and MDD patients [33,69,148]. 

### 7.5. Linear and Logistic Regression

A Linear Regression model is a supervised regression model that predicts a continuous output. It finds the optimal hyperplane that minimizes the sum of squared difference between the true data points and the hyperplane. A Logistic Regression model is a supervised classification model that can be used for binomial, multinominal, and ordinal classification tasks. Logistic Regression classifies observations by examining the outcome variables on the extreme ends and determines a logistic line that divides two or more classes [136]. Linear and Logistic Regression are popular in algorithms as they are easy to implement, efficient to train, and easy to interpret. However, a limitation of both models is that they make multiple assumptions, e.g., that a solution is linear, the input residuals are normally distributed, and that all features are mutually independent [150]. Multicollinearity, the correlation between multiple features, and outliers will inflate the standard error of the model and may undermine the significance of significant features [151]. Further, outliers that deviate from the expected range of the data can skew the extreme bounds of the probability, making both algorithms sensitive to outliers in the dataset [150].

Linear Regression has been used to quantify tremors among Essential Tremor (ET) patients [116] and to estimate motor-related symptom severity among PD patients [31,93]. It has also been used to forecast convergence between body sides for Hemiparetic patients [130]. Logistic Regression was a popular approach for classifying PD diagnosis [107,110], Post-Traumatic Stress Disorder [109], and distinguishing fallers and non-fallers [152]. Logistic Regression has been used to classify drug effects, such as predicting the pre- and post-medication states among PD patients [22]. 

### 7.6. Neural Networks

Neural Networks (NN), also known as Artificial Neural Networks (ANN), can be used for unsupervised and supervised classification and regression tasks [153]. NN consists of a collection of artificial neurons (or nodes). Each artificial neuron receives, processes, and sends the signal to the artificial neuron connected to it. The neurons are aggregated into multiple layers, and each layer performs different transformations on the signal. The signal first travels from the input layer into the output layer while possibly traversing multiple hidden layers in between. NN offer several advantages, such as the ability to detect complex non-linear relationships between features and outcomes and work with missing data, while it also requires less preprocessing of the data and offers the availability of multiple training algorithms. However, the disadvantages of NN include increased computational burden, reduced explainability and interpretability (as NN are ‘black box’ in nature), and the fact that NN are prone to overfitting [154]. However, it is important to highlight the growing number of studies that specifically explore explainable deep learning approaches for biomarker discovery and development. Studies utilizing methodologies such as LIME (Lime Tabular Explainer), SHAP (Shapley Additive exPlanations), and other visual inspections of feature distribution and importance have aided clinicians in understanding the model mechanisms. These approaches also provide patient-specific insights by describing the importance of each feature, which may, in turn, facilitate personalized treatment opportunities [90,155,156,157]. 

The most popular applications for neural networks were for the classification of a diagnosis or classification of a state or event. The most popular application is the detection of tremors among PD patients [23,52,86,137,158]. NN have been used to classify unipolar and bipolar depressed patients based on motor activity [45,159], estimate depression severity [159], forecast seizures [160], and classify a treatment response using keyboard patterns among PD patients [161].

### 7.7. Transfer Learning

Transfer learning (also known as domain adaption) refers to the act of deriving the representations of a previously trained ML model to extract meaningful features from another dataset for an inter-related task [162]. One applicable scenario is the training of a supervised ML model on data collected in a controlled setting (such as in a lab or clinic). The performance of the model may suffer when applied to a dataset collected under free-living conditions. Rather than developing a new model trained solely on a free-living condition dataset, transfer learning can use patterns learned from the controlled setting dataset to improve the learning of the patterns from the free-living conditions dataset. 

Transfer learning can also be a valuable technique for enhancing the utilization of limited or rare data [163]. One practical application is to employ pretraining on abundant control data and subsequently finetune the model on the specific population of interest to improve the model’s performance [163,164,165]. This approach not only optimizes the efficiency of utilizing scarce data but also facilitates model personalization. By adapting a pretrained model to individual characteristics or preferences, it becomes possible to create personalized models that better cater to unique needs or circumstances. Transfer learning thus offers a powerful means to leverage existing knowledge and make the most of available data resources, enhancing both the efficiency and personalization of biomarkers.

Given its application, transfer learning reduces the amount of labeled data and computational resources required to train new ML models [162], thus making this method advantageous when the sensor modalities, sensor placements, and populations differ between studies. While we only identified two studies that applied transfer learning to estimate PD disease severity using movement sensor data [166,167], we predict that the application of transfer learning will enable future researchers to overcome the challenges of a limited dataset and develop more sensitive and effective ML models. 

### 7.8. Multi-Task Learning

Multi-task learning (MTL) enables the learning of multiple tasks simultaneously [168]. Learning the commonalities and differences between multiple tasks can improve both the learning efficiency and the prediction accuracy of the ML models [168]. A traditional single-task ML model can have a performance ceiling effect, given the limitations of the dataset size and the model’s ability to learn meaningful representations. MTL uses all available data across multiple datasets and can learn to develop generalized models that are applicable to multiple tasks. To use MTL, there should be some degree of information shared between or across all tasks. The correlation allows MTL to exploit the underlying shared information or principles within tasks. Sometimes MTL models can perform worse than single-task models because of ‘negative transfers’. This occurs when different tasks share no mutual information or if the information of tasks are contradictory [169]. MTL models have been used to simultaneously model data sourced from two separate sources or to model multiple outcomes [170,171]. For example, Lu et al. explored the use of MTL to jointly model data collected from two different smartphone platforms (iPhone and Android) to jointly predict two different types of depression assessments (QIDS and a DSM-5 survey) [79]. They illustrated that the classification accuracy of the MTL approach outperformed the single-task learning approach by 48%; thus, the classification model benefited from learning from observations sourced from multiple devices. 

### 7.9. Generalized vs. Personalized 

ML algorithms can be trained on population data or individual subject data. Generalized models, which are trained on population data, are fed data from all participants for the purpose of general knowledge learning. Conversely, personalized models are trained on an individual’s data and take into consideration individual factors such as biological or lifestyle-related variations [172]. We have adopted these terms from Kahdemi et al.’s study, in which they developed generalized and personalized models for sleep-wake prediction [173]. The heterogenous nature of each population or individual can be a potential hinderance for generalizable models. A single individual’s deviation from the ‘norm’ may be viewed as a source of ‘noise’ in a generalized model. For example, patients with mood disorders such as MDD and BD have large inter-individual symptom variability. Abdullah et al., reliably predicted the social rhythms of BD patients with personalized models using smartphone activity data [30]. Cho et al. compared the mood prediction accuracy of personalized and generalized models based on the circadian rhythms of MDD and BD participants [38]. Their studies illustrated that their personalized model predictions were, on average, 24% more accurate than the generalized models. These studies lay the groundwork for developing personalized models that are more sensitive to individual differences.

## 8. Model Hyperparameters

The process of building an effective ML model consists of two main steps: selecting the appropriate ML algorithm and optimizing the model performance by tuning its parameters. Each model consists of two types of parameters: (i) the parameters that are initialized and continuously updated throughout the learning process (e.g., the weights of neurons of a neural networks) and (ii) the hyperparameters that must be set prior to the learning process as they define the model architecture (e.g., the regularization parameters of a Linear Regression model, and the learning rates of a neural network) [174]. Every combination of the selected hyperparameters will have a direct influence on the performance of the learned model. For example, as the number of trees in a RF increases, the more features tend to be selected by the model, which may not always be relevant for the development of biomarkers [175]. Similarly, the number of layers, number of neurons per layer, activation functions, and the regularization techniques used for NN can each influence the model performance [176]. While most ML algorithms come with default values for the hyperparameters, these may not be optimal for the dataset at hand, and even tuned hypermeters are at risk of being non-optimal for a different dataset. The process of selecting the optimal hyperparameter configurations is known as hyperparameter tuning [177]. 

To identify the optimal hyperparameters for a model, researchers must define the hyperparameter space and the hyperparameter search strategy. When defining the hyperparameter space, the distribution of the hyperparameter ranges can be either uniform or logarithmic. The uniform distribution assigns equal probability to all hyperparameter values within a manually defined range. The log-uniform distribution samples hyperparameter values uniformly between the logarithmic transformations of the lower and upper thresholds. We argue that log-uniform distribution is particularly useful when exploring values that vary over several orders of magnitude. Consider the example of tuning a linear regression model with the hyperparameter alpha, which determines the strength of regularization. To efficiently explore a wide range of alpha values, such as between 0.001 and 10, the log-uniform distribution allows for an evenly distributed search space over different orders of magnitude. Log-uniform distribution can be used for the initial exploration of a large range of hyperparameter values. The range can then be narrowed down to explore with a uniform-distribution to determine the optimal hyperparameters for the respective models. 

The manual tuning of hyperparameters is impractical due to the large number of available hyperparameters, hyperparameter configurations, and time-consuming model evaluations. Automated tuning approaches are preferred, and there are a wide variety of approaches available, including GridSearch, RandomSearch, and Bayesian Optimization [177]. GridSearch uses brute force to test a finite combination of hyperparameters to identify the optimal hyperparameter configuration [178]. This approach can suffer from the effects of dimensionality, as more potential hyperparameter configurations can be time-consuming and computationally expensive. An alternative to GridSearch is RandomSearch. RandomSearch only samples a subset of all possible hyperparameter configurations within a specific time or computational budget [179]. While RandomSearch only relies on a subsample of configurations, it has been shown to outperform the GridSearch method [179]. As GridSearch and RandomSearch do not consider previous performance evaluations for their hyperparameter optimization strategy, they are inefficient in exploring the hyperparameter search space. Bayesian Optimization, which uses Bayes Theorem, is a powerful approach. It considers previous hyperparameter evaluations to choose which hyperparameters to evaluate next and disregards potential hyperparameter combinations that are deemed irrelevant [178]. This approach reduces the time and computations required for hyperparameter tuning. The benefit of using these more automated approaches to hyperparameter tuning is three-fold. First, it reduces the time effort required to optimize a ML model. Next, the performance of the ML models is improved as the hyperparameters explore different optimal model configurations for different datasets. Finally, when the hyperparameters and their ranges (together also referred to as the hyperparameter space) and the hyperparameter tuning methods are reported, the models and the findings become reproducible [180]. When similar hyperparameter tuning processes can be used for different ML algorithms for different datasets, researchers can then identify the optimal ML model. 

Among the selected studies, 25 discussed which hyperparameters were considered for their models [23,24,34,43,44,46,53,69,73,86,87,94,95,107,108,109,110,114,138,158,159,181,182,183,184], of which one stated they used the default hyperparameters of the models [69]. Only nine studies discussed how they selected or optimized their hyperparameters. We identified four studies that stated GridSearch was used for the hyperparameter tuning [36,46,95,110]. We did not identify any studies that used RandomSearch or Bayesian Optimization. The limited reporting of hyperparameters and the hyperparameter tuning process poses a problem for the transparency, reproducibility, and comparison of ML models.

## 9. Model Evaluation

Assessing a ML model’s performance is an essential component for determining the usability and reliability of the model. Depending on the objective of the research, it is often necessary to try to compare the performance of multiple ML models to identify the optimal model [185,186]. In ML, the terms metric and measure are often used interchangeably, but they do have slightly different meanings. A metric is a function used to evaluate the performance of a model, while a measure is a numerical summary of the performance of a model obtained using one or more metrics. It is best practice to use multiple metrics and model performance visualizations for the model evaluation, as a model may perform well for one evaluation metric and poorly for another. Using multiple evaluation metrics ensures that the model is operating optimally and correctly. The following sections provide more details about the performance metrics used for classification and regression models. Table 4 provides an overview of the most common performance metrics used in the selected studies, their respective calculations, and their clinical interpretations.

### 9.1. Classification Measures

Classification models have discrete outcomes; thus, a metric must reflect how often an observation belongs to the correct label or class [187]. There are three categories of classification measures: Threshold Metrics, Ranking Metrics, and Error Metrics. Threshold Metrics (such as accuracy and F1 score) quantify the prediction errors of the classification model as a ratio or rate. Ranking Metrics (such as the Receiver Operating Characteristics (ROC) and Area Under the Curve (AUC)) focus on evaluating classification models based on how effective they can discern separate classes. Error Metrics (such as Root Mean Square Error) quantify the uncertainty of the classification model’s predictions. While the Threshold and Ranking Metrics are focused on correct and incorrect predictions, the Error Metrics quantify the proportion of classification errors.

As ML models are increasingly being used to perform high-impact tasks pertaining to clinical assessments, an evaluation metric must be selected based on what the stakeholders find to be important regarding the model prediction, which can make the selection of the model metrics challenging. As seen in Table 4, accuracy, sensitivity, specificity, and precision are calculated based on four test results. The True Positive (TP) and True Negative (TN) indicate the presence or absence of a diagnostic or characteristic. The False Positive (FP) and False Negative (FN) indicate the opposite of the true condition. 

Binary classification models typically involve a decision threshold hyperparameter that determines how the model assigns labels based on the predicted probabilities. The default threshold is typically 0.5, meaning that if the predicted probability is greater than 0.5, the positive label is assigned, and vice versa. However, it is important to note that this threshold can be adjusted to accommodate specific needs or domain considerations. To evaluate the performance of binary classification models across different decision thresholds, the ROC curve is commonly used. The ROC curve provides an overview of the model’s performance by illustrating the trade-off between TP and FP rates at various threshold values. ROC can aid the assessment of the model’s performance across a range of decision thresholds and enable the selection of the threshold that aligns with a specific objective.

It is worth noting that many classification metrics, including accuracy, precision, recall, and F1 score, assume binary labels. However, when dealing with multiclass classification problems, another approach is to use one-vs-rest or one-vs-one strategies, wherein the problem is decomposed into multiple binary classification tasks. The performance of the model on each task can then be evaluated using the binary classification metrics, and the results can be aggregated or averaged to provide an overall assessment of the model’s performance on the multiclass problem.

Class imbalance can be an obstacle for assessing model performance. In particular, accuracy, AUC, ROC, may be sensitive to such imbalances [188]. Hence, when facing class imbalance, there are two approaches to consider: one can choose a metric that accounts for class imbalance or one can choose to balance the classes. Metrics such as balanced accuracy, F1-score, or Matthews Correlation Coefficient (MCC) are common metrics for handling class imbalance, as identified by 15 studies [23,24,29,36,44,60,61,107,108,110,114,140,159,161,189]. Balanced accuracy represents the mean of the sensitivity and specificity, while the F1-score represents the mean of the precision and recall [190]. The MCC measures the correlation coefficient of the binary and even multiclass classes. Therefore, the MCC score is high only if the classification model correctly predicts both the positive and negative predictions [190,191]. 

The other approach to handling class imbalances is adjusting the class distribution using oversampling or undersampling. We identified eight studies that used random over/under sampling or SMOTE [29,44,45,46,61,95,109,192]. Oversampling techniques duplicate the samples of the minority class, while undersampling removes samples of the majority class. However, these techniques also have their disadvantages, as the duplication of multiple samples can lead to overfitting of a model, while undersampling reduces the diverse representation of the majority class. Thus, we would specifically recommend using the Synthetic Minority Oversampling Technique (SMOTE) with Tomek Links or Edited Nearest Neighbor (ENN)—two undersampling techniques [193,194]. SMOTE is first applied to create an artificial minority class to minimize the class imbalance. Next, Tomek Links or ENN can be used to remove samples that are close to the boundaries between the classes, which would further separate the classes [193,194].

### 9.2. Regression Measures

As regression models generate predictions on a continuous scale, the objective is to estimate how close the predictions were to the true values [195]. Among the studies selected, we found that regression models used Distance Metrics and Error Metrics to estimate the strength of the association or the distance between the predicted values and the true values [29,42,87,93,96,128,152]. We would like to emphasize that these metrics are used to compare the performance of the composite biomarkers rather than the performance of the individual features. The most common Distance Metrics were the correlation (also known as R) and the percentage of the variance explained (R^2^). Both were used to assess the strength of the association between the predicted and true values [196]. There is no rule of thumb for interpreting the strength of R^2^. While an R^2^ closer to 1 can be obtained in clinical trials, a low R^2^ can still be useful with respect to trends in the data. We would like to address two points of caution when using the R^2^ [185,187]. First, it is not always suitable to compare R^2^ across different datasets, as different clinical populations are likely to differ in their feature variance. Second, the R^2^ will increase with the number of features. To compensate for this, one may use the adjusted R^2^ to account for the number of features [197,198]. 

The Error Metrics included the Mean Absolute Error (MAE), Mean Squared Error (MSE), and Root Mean Squared Error (RMSE) [133]. The MAE measures the average absolute difference between the true and predicted values. The MAE is easy to interpret and robust to outliers. The absolute difference accounts for negative differences. The MSE squares the error instead of providing the absolute error, which gives more weight to the bigger errors. The MSE is sensitive to outliers and not easy to interpret, as the results will not have the same unit as the output. However, the RMSE provides an estimation of the error in the same units as the output while maintaining the properties of the MSE [199].

## 10. Model Validation

In ML, model validation refers to the process of evaluating the generalizability of a trained model on an unseen dataset. Selecting the most appropriate model validation approach depends on the size and characteristics of the datasets. Three datasets are required for model validation: the training, test, and validation datasets. In most cases, the validation dataset can be a subset of the original dataset; however, this can lead to data leakage, which could produce overly optimistic results. Another approach is to create a validation dataset from an independent (but comparable) dataset, which ensures an unbiased and independent evaluation of the ML model. However, a limitation is that the performance evaluation may reflect high variance due to the limited size of the dataset [200]. Moreover, it is crucial to highlight that a participant should only be present in a single dataset, such as the training dataset, and should not simultaneously appear in other datasets such as the testing or validation datasets. When a participant’s observations are distributed across multiple datasets, data leakage can occur, compromising the accuracy estimation and its applicability to new participants [183]. As a result, cross-validation on the observation level rather than the participant level is methodologically flawed. Unfortunately, this is a common issue and needs to be accounted for in future studies [201]. 

Cross-validation is a popular validation method that uses resampling to train, test, and validate a model using different subsets of the data. The training dataset is used to train the ML model to learn the patterns within a dataset. The validation dataset is used to tune the hyperparameters of the model based on the performance of the ML model trained on the training dataset. The test dataset provides an unbiased estimate of the performance of the final ML model after training and validation. In the scenario when both validation and test datasets are used, the test datasets are only used to assess the model once (via hold-out validation) or multiple times (via nested cross-validation). In general, datasets need to meet two main requirements. The datasets should not have shared or overlapping observations to ensure that data leakage does not lead to bias in the estimates, and all observations must be statistically independent [202]. When applying feature engineering or feature selection with cross-validation, any transformation or selection steps should be performed within each fold of the cross-validation to prevent biasing in the training of the prediction model with information from the test dataset [203]. The overall performance of the prediction models, obtained by averaging across each iteration of the cross-validation, evaluates the effectiveness of the combined feature reduction and learning methods in estimating the label for a given dataset.

Among the selected studies, we found that the most popular cross-validation methods were k-fold cross-validation (N = 27), Leave-One-Out cross-validation (N = 16), and custom validation (N = 8). Overall, 15 studies did not report the use a validation method. K-fold cross-validation randomly splits the datasets in ‘k’ folds; one-fold is used for testing and the remaining folds are used for training. This step is repeated until every unique fold has been used as the test dataset, and the overall performance is based on the average of the performance of each model in each fold [204]. Leave-one-out cross-validation is a specific type of k-fold cross-validation, wherein individual observations (or participants) are the test datasets, and the remaining cases are used for training. Leave-one-out cross validation prevents data leakage across datasets, as repeated measurements of the same subjects can lead to the violation of independence assumption for ordinary cross-validation [204,205,206]. 

We would like to highlight the advantages of the nested cross-validation approach. While nested cross-validation was the least popular approach, we would argue that nested cross-validation is a more robust approach for selecting and evaluating a ML model [207]. Currently, the model section without the nested cross-validation approach uses the same data to both tune the model hyperparameters and evaluate its performance. Therefore, information is “leaked” between the training and validation of the model, which can lead to overfitting [207]. Nested cross-validation consists of an inner loop and an outer loop. The outer loop assesses the model performance, while the inner loop assesses the hyperparameter selection [207]. Each iteration of the outer loop is split into a different combination of training and test sets. The outer loop training set is used in the inner loop, which is further split into a training and validation dataset. The inner loop split is repeated over k-folds, and the best performing model across the k-folds is evaluated in the outer loop. This ensures that different data are used to optimize the models’ hyperparameters and evaluate the model’s performance. The final model performance represents the average and standard deviation of the model performance as selected by each of the outer loops. Without the standard deviation or confidence intervals, it is not possible to evaluate the spread or stability of the prediction error of the given models [208,209].

It is important to highlight that cross-validation is only used to approximate the generalization error of the models built and not to build the final model that will be used for making predictions [205,210]. The average prediction error across the folds gives an expected error for a single model built on the single dataset. If the variance of the prediction error is too high, then the model is considered unstable. To select a single model, it is recommended that researchers rebuild the model using the full dataset [211]. If an external validation set is available, then this validation set can be used to evaluate and compare the single prediction error to that of the cross-validation prediction error.

## 11. Recommendations

In this recommendation section, we address the main issues consistently identified in the selected studies and how to amend these issues for future trials (see Figure 5 for a simplified overview of these recommendations). It is important to bear in mind the regulatory implications for developing ML-derived biomarkers. Within the European Union, AI medical systems and devices are considered high risk; therefore, they are subject to stringent reviews prior to being made available on the market [212]. These review requirements emphasize the importance of achieving high levels of performance, transparency, and minimal risk in ML-derived biomarker development [213]. High performance implies that the developed ML models must be accurate, robust, and capable of reliably and consistently predicting the target outcome variable. Furthermore, transparency in ML-derived biomarker development refers to the provision of clear and adequate information to the user, including appropriate human-readable measures to minimize risks associated with the use of the system. The development of ML-derived biomarkers must also aim to minimize risks and discriminatory outcomes, which can be achieved by training the ML model on high-quality datasets that are representative of the target population and by conducting adequate risk assessment checks [214]. These considerations are critical for ensuring the safe and effective use of ML-derived biomarkers in clinical practice.

### 11.1. Inclusion of Healthy Controls

When conducting a study focused on disease classification or estimation, the inclusion of control data can serve several purposes. By comparing the data from individuals with the condition of that of the healthy controls, researchers can discern whether the observed differences are specific to the condition or a result of unrelated factors. Moreover, analyzing the performance of a model on control subjects can shed light on the biomarker’s effectiveness and reliability. By evaluating how well the model distinguishes between healthy controls and patients with the condition, researchers can gain a better understanding of its predictive capabilities. This evaluation can provide insights into potential false positives or false negatives that may occur when using the model in real-world settings.

It is worth noting that, when including control data, the control data should be appropriately matched with the patient population data. Having age- and gender-matched control subjects can help minimize confounding variables, improving the accuracy of the analysis. This matching process allows researchers to draw more robust conclusions about the relationship between the identified features or patterns and the disease activity while also reducing the potential impact of demographic factors on the results.

The finding that only half of the studies included healthy controls is significant as it highlights a potential gap or limitation in the existing body of research. Without the inclusion of controls, it becomes challenging to attribute identified features or patterns solely to the CNS disorder or the severity of the condition. Further, if the dataset only contains a relatively homogeneous population, it calls the reliability and predictive capabilities of the models into question. We encourage future researchers to include control subjects in their studies, as it would improve the strength of their biomarkers and the validity of their findings.

### 11.2. Data Quality and Preprocessing

The remote monitoring of clinical trials can generate large and complex datasets that include longitudinal data from multiple subjects and data sourced from multiple sensors, resulting in a multi-dimensional data structure. To this point, we recommend using the WHO mHealth Technical Evidence Review Groups’ mHealth evidence and evidence reporting and assessment (mERA) 16-item checklist to provide transparency on which mHealth invention was used, where, and how it was implemented to support the reproducibility of the mHealth data collection [215]. To ensure the quality and reliability of the data, it is important to assess the quality of the data. This assessment includes examining the data for missing and outlier data and understanding how these factors might affect the generalizability and reproducibility of the ML model. While most studies provide detailed information on patient populations, the devices used, and the data collected, they often underreport information related to data quality and preprocessing steps. Therefore, it is important to provide sufficient details on the methods used to preprocess the data, including the quantity of missing and outlier data and the strategies employed to handle such data. This information can ensure that the data collection and preprocessing process can be reproduced, which, in turn, can enhance the credibility and generalizability of the ML model.

### 11.3. Feature Engineering and Selection

There is a wide variety of manual or automated techniques used for engineering and selecting features to feed a model. ML models perform best when feature engineering and selection are leveraged to formulate potentially clinically relevant features from existing data. In addition, the performance of the ML model can be optimized, and the computational time can be reduced when the redundancy across the features is reduced. While only selecting the most informative features can remove noise (therefore reducing the likelihood of overfitting), selecting too few features may reduce the strength of the (combined) signal in the dataset, making the ML model vulnerable to underfitting. Feature engineering and selection can be guided by domain expertise and/or automated statistical models, where multiple features are evaluated by their importance in predicting the outcome. While automated feature engineering techniques, such as clustering, PCA, and DL, can be used to extract a reduced set of representative features, this risks a potential decline in interpretability, which may limit its clinical application.

### 11.4. Model Configuration and Optimization

When selecting the ML models, there are several factors that should be considered, such as model objectives, model types, model hyperparameters, and model evaluation. Poor design choices and lenient hyperparameter tuning and validation in these steps can lead to poor model performance. We recommend that researchers carefully consider each step of building their ML pipeline by comparing multiple ML algorithms, using automated methods for assessing multiple hyperparameter configurations, and using nested cross-validation to both optimize and validate the ML models.

### 11.5. Model Validation

We would recommend using a minimum of three datasets to validate a ML model and train, validate, and test a dataset. At no point should the test set be used for the model configuration, which includes the data transformation, feature engineering, and selection, or the tuning of the hyperparameters. The test dataset could either be a subset of the original data (with no overlapping subjects or observations) or a separate external dataset. The use of an external dataset is ideal as this ensures that there is no influence of bias during the data collection period and that there is no data leakage between the datasets. If an external dataset is not available or if the dataset is not sufficiently large, we recommend nested cross-validation. This resampling method supports model hyperparameter tuning and performance evaluation without the risk of data leakage across the dataset.

It is crucial to report the evaluation metric results for each dataset. In the case of cross-validation reporting, we recommend that researchers report the distribution of the performance measures (e.g., the mean and standard deviation or median and 95% confidence interval) across the folds to show the average and variability of the performance of the models. As cross-validation evaluates the prediction error across multiple ML models, we would also recommend reporting the performance of the final model selected. This is achieved by re-training a ML model on the full dataset and evaluating the performance on an external dataset [207,210]. This would give insight into how well the model would perform under different circumstances. We also highly recommend using multiple evaluation metrics for assessing the model’s performance. Seeing as a model might excel for one metric and fail for another, this underscores the need for comprehensive evaluation. Employing multiple metrics ensures optimal operation and reduces the likelihood of blind spots.

Once the final model has been trained, there are three approaches to choose from to apply the model to a new target dataset. The first approach is to test the model “as-is”, implying that the ready-made model can be used in its original state without modifications [216]. In the second scenario, the train data and the target data may have different characteristics, which may lead to a distribution shift. The type of distribution shift between the two datasets can occur for many reasons, including different mHealth devices used for data collection, environmental noise, and sampling bias [217]. When this occurs, transfer learning can be used to finetune the ready-made model and update its weights to better suit the target dataset [216]. In the third scenario, the target dataset may have different requirements than the original training dataset [216]. As a result, the decision boundary of the classification model can be altered, such as optimizing the model for a sensitivity of 90% instead of accuracy. Whether testing the model as-is, employing transfer learning, or adjusting the decision boundary, these strategies offer flexibility in adapting the model to different settings and improving its performance for validation purposes.

### 11.6. Model Reproducibility and Interpretability

Equally important as the model performance are the ML models’ reproducibility and interpretability. Reproducibility is a core component for ensuring that a ML model can be validated and reused by clinical researchers. Technical reproducibility involves using the same computational procedures to produce consistent model outcomes. Statistical reproducibility ensures that the model demonstrates similar statistical performance across different subsets of data. Conceptual reproducibility refers to achieving consistent results under new conditions, such as data collected from different settings [216]. Transparency regarding data quality, feature engineering and selection methods, the hyperparameters considered and selected, and the model validation protocol can help ease the ability of the scientific community to recreate the work in the published literature. Best practices for reproducibility include publishing the code on GitHub or by publishing FAIR metadata [211,218,219].

Given the potential clinical application of ML models, prior to modeling, researchers should determine the model’s interpretability requirement. While ML models provide researchers with what was predicted, interpretability requires that the model can explain why it made the prediction [185]. Interpretability enables us to understand the causal relationships between the data and the ML model’s predictions. There are two situations in which the interpretability of a model is required: when an inaccurate prediction can have severe or even fatal consequences for the patients (such as a misclassified diagnosis [220]) and when the interpretability can be used to identify novel relationships between clinical factors and the predicted outcome (such as factors influencing treatment outcomes [221]). There can be two situations in which interpretability is not required: situations in which incorrect predictions do not have severe consequences (such as counting the number of coughs [222]) or situations in which the ML model has been sufficiently validated in real clinical applications, even if the predictions are not perfect [223]. While black box models may offer more accurate predictions than an interpretable model, they only provide limited insight into how the predictions were made. Therefore, both interpretable and black box models have their respective merits.

There are two broad approaches towards achieving interpretability [224]. One approach is to use easy-to-interpret models, such as Linear or Logistic Regression, where the coefficients of the features can provide insight into the features’ associations with the predicted outcome. The other approach is to use explanation methods for explaining complex or black box models, such as SHapley Additive exPlanations plots (SHAP), Local Interpretable Model-agnostic Explanations (LIME), or Anchors [224]. We recommend that researchers report whether their final selected model was an interpretable model or a black box [225]. If it was interpretable, we recommend discussing what interpretations can be derived from the models.

## 12. Conclusions

The rise and breadth of ML applications in clinical trials highlight the increasing reliance and importance of ML in the development of novel biomarkers [226]. While the advances in ML applications have demonstrated great potential for innovative biomarker development, the process of its development is not well documented, which, in turn, limits the reproducibility of these findings. This review has illustrated the steps taken to translate raw data from mHealth technologies into meaningful clinical biomarkers using ML. Given the lack of consistent reporting in the ML methods, the present review cannot provide a complete or detailed picture of the notable and generic practices. However, the authors have provided an overview of the status quo of the development and translation of ML-derived biomarkers in mHealth-focused clinical trials. The recommended checklist provided in the review could serve as a foundation for the design of future ML-derived biomarkers in conventional ML practices. By encouraging consistent and transparent reporting, researchers can accelerate the integration of novel biomarkers derived from mHealth sensors and ML pipelines into future clinical trials.

## Figures and Tables

**Figure 1 sensors-23-05243-f001:**
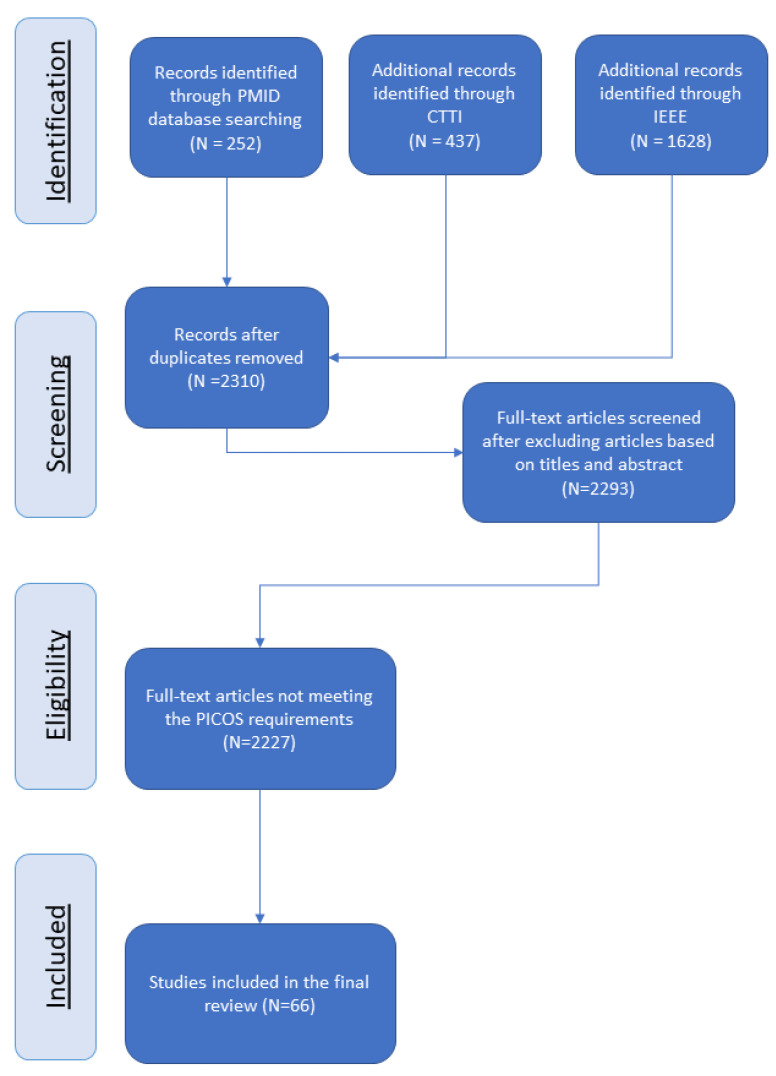
Flow diagram illustrating the paper selection process for this review.

**Figure 2 sensors-23-05243-f002:**
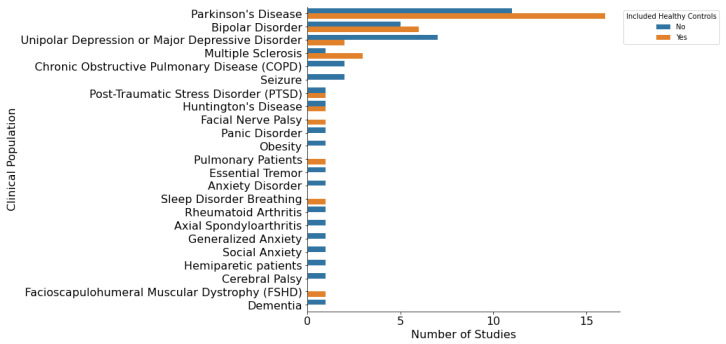
Clinical populations and the use of healthy controls in the selected studies.

**Figure 3 sensors-23-05243-f003:**
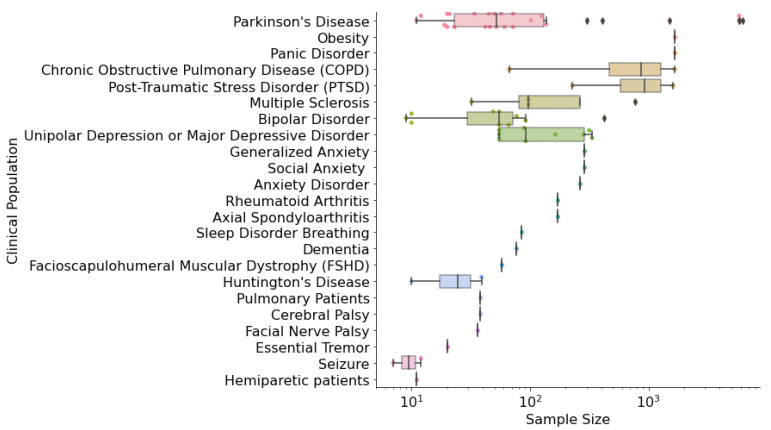
Sample sizes of clinical populations included in selected studies, with x-axis (sample size) presented on a logarithmic scale.

**Figure 4 sensors-23-05243-f004:**
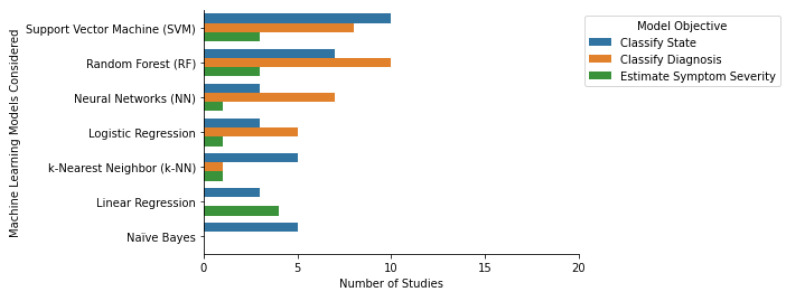
Machine learning algorithms and their respective objectives in the selected studies.

**Figure 5 sensors-23-05243-f005:**
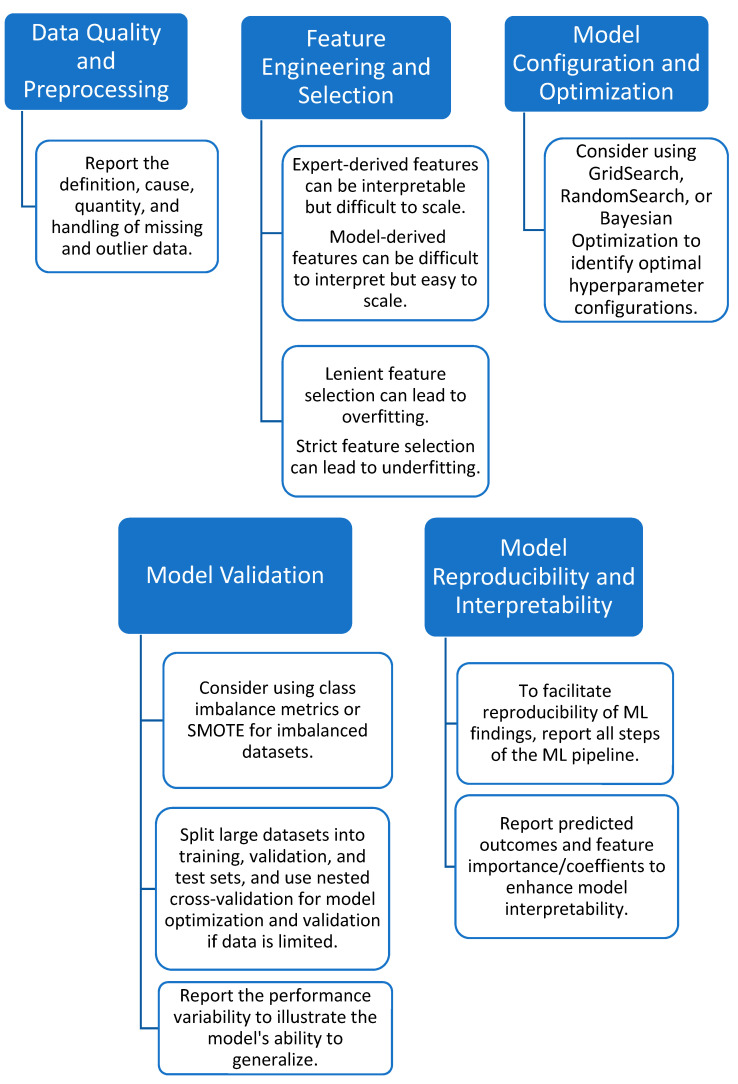
General recommendations for building an effective and reproducible ML pipeline.

**Table 1 sensors-23-05243-t001:** Representation of a standard machine learning pipeline.

Stage	Objective	Example
Study Design	The ML pipeline is provided with a study objective in which the features and corresponding outputs are defined. The ML model aims to identify the associations between the features and outputs.	The study objective is to classify Parkinson’s Disease patients and control groups using smartphone-based features.
2. Data Preprocessing	Data preprocessing filters and transforms raw data to guarantee or enhance the ML training process.	To improve the model performance, one may identify and exclude any missing or outlier data.
3. Feature Engineering and Selection	Feature engineering uses raw data to create new features that are not readily available in the dataset. Feature selection selects the most relevant features for the model objective by removing redundant or noisy features. Together, the goal is to simplify and accelerate the computational process while also improving the model process. For deep learning methods, the concept of “feature engineering” is typically embedded within the model architecture and training process, although substantial preprocessing steps may occur prior to that.	An interaction of two or more predictors (such as a ratio or product) or re-representation of a predictor are examples of feature engineering. Removing highly correlated or non-informative features are examples of feature selection. Note: The feature selection step can occur during model training
4. Model Training and Validation	During training, the ML model(s) iterates through all the examples in the training dataset and optimizes the parameters of the mathematical function to minimize the prediction error. To evaluate the performance of the trained ML model, the predictions of an unseen test set are compared with a known ground truth label.	Cross-validation can be used to optimize and evaluate model performance. Classification models may be evaluated based on their prediction accuracy, sensitivity, and specificity, while regression models may be evaluated using variance explained (R2) and Mean Absolute Error.

**Table 2 sensors-23-05243-t002:** An overview of the keyword strategy used for this study.

Domain	Search String
Technology	((“smartphone”[tiab] OR “wearable”[tiab] OR “remote + monitoring”[tiab] OR “home + monitoring”[tiab] OR “mobile + sensors”[tiab] OR “mobile + montoring”[tiab] OR “behavioral + sensing”[tiab] OR “geolocation”[tiab] OR “mHealth”[tiab] OR “passive + monitoring”[tiab] OR “digital + phenotype”[tiab] OR “digital + phenotyping”[tiab] OR “digital + biomarker”[tiab])
Analysis	AND (“machine + learning”[tiab] OR “deep + learning”[tiab] OR “random + forest”[tiab] OR “neural + network”[tiab] OR “time + series”[tiab] OR “regression”[tiab] OR “SVM”[tiab] OR “knn”[tiab] OR “dynamics + model”[tiab] OR “decision + tree”[tiab] OR “discriminant + analysis”[tiab] OR “feature + engineering”[tiab] OR “feature + selection”[tiab] OR “data + mining”[tiab] OR “model”[tiab] OR “classification”[tiab] OR “diagnostic”[tiab] OR “prognostic”[tiab] OR “symptom + severity”[tiab] OR “prediction”[tiab] OR “monitoring”[tiab])
Population	AND (“disease”[tiab] OR “disorder”[tiab] OR “diagnosis”[tiab] OR “prognosis” OR “alzheimer”[tiab] OR “parkinson”[tiab] OR “Huntington”[tiab] OR “neurodegenerative”[tiab] OR “degenerative” OR “tremor”[tiab] OR “bipolar”[tiab] OR “depression”[tiab] OR “manic”[tiab] OR “anxiety”[tiab] OR “vocal + biomarker”[tiab] OR “amyotrophic + lateral + sclerosis”[tiab] OR “central + nervous + system”[tiab] OR “symptom”[tiab] OR “psychosis”[tiab] OR “stroke”[tiab] OR “muscular dystrophy”[tiab] OR “Facioscapulohumeral Dystrophy”[tiab] OR “autoimmune”[tiab] OR “seizure”[tiab] OR “multiple + sclerosis”[tiab])
Date	AND (“2012/01/01”[PDAT]:”2022/12/31”[PDAT])
Language	AND (English[lang])
Exclusion Criteria	NOT(“animals”[tiab] OR “implant”[tiab] OR “hospital”[tiab] OR “caregiver”[tiab] OR “telemedicine”[tiab] OR “telerehabilitation”[tiab] OR “smartphone + addiction”[tiab] OR “nursing”[tiab] OR”screening”[tiab] OR “recruitment”[tiab] OR “diabetes”[tiab] OR “malaria”[tiab] OR “self-care”[tiab] OR “self-management”[tiab] OR “self-help”[tiab])
Article Type	AND (clinicalstudy[Filter] OR clinicaltrial[Filter] OR clinicaltrialphasei[Filter] OR clinicaltrialphaseii[Filter] OR clinicaltrialphaseiii[Filter] OR clinicaltrialphaseiv[Filter] OR controlledclinicaltrial[Filter] OR meta-analysis[Filter] observationalstudy[Filter] OR randomizedcontrolledtrial[Filter] OR systematicreview[Filter])

**Table 3 sensors-23-05243-t003:** Table of the inclusion and exclusion criteria used for study selection.

Category	Criteria
Population	The study must be initiated by a research organization and not by the participants. The participants must have a clinical diagnosis that is affected by the CNS. Hence, studies that collected data from participants with no clinically confirmed diagnosis were not considered.
Intervention	The study must include the use of smartphone or non-invasive wearables to remotely monitor and quantify passive biomarkers under free-living conditions.
Comparator	A ground truth comparator for digital phenotyping such as clinical assessment, medical records, or self-reported outcomes.
Outcomes	A ML model that is used to classify a clinical label (such as a diagnosis, or clinical event), estimate symptom severity, or to detect treatment effects.
Study Type	The paper must be about a human-centered observational study (cohort or longitudinal) where the data were collected outside the clinic, lab, or hospital (free-living conditions). Hence, studies that use smartphones or wearables as a form of intervention or as screening tools are not of interest. The study must show if the ML models had ecological validity by validating the models using free-living data. The study has to have been written or translated into English and published within the last 10 years (2012 onwards).

**Table 4 sensors-23-05243-t004:** Clinical interpretations of common ML performance metrics (True Positive = TP, True Negative = TN, False Positives = FP, False Negatives = FN, Sum of Squares of Residuals = RSS, Total Sum of Squares = TSS, Number of Observations = N).

Term	Title 2	Title 3
Accuracy	$\frac{TP+TN}{All\;Observations}$	Out of all the predictions, how many predictions were correctly identified as positive or negative?
Precision	$\frac{TP}{TP+FP}$	How many predictions were correctly labeled as patients out of all correctly classified patients and misclassified healthy controls?
Specificity	$\frac{TN}{TN+FP}$	How many predictions were correctly labeled as healthy controls out of all healthy controls? In other words, of all healthy controls, who were correctly identified as such?
Recall/Sensitivity	$\frac{TP}{TP+FN}$	Of all the patients, who were correctly classified/identified as such?
F1-score	$\frac{2x(Precision\;\times \;Recall)}{Precision+Recall}$	How many predictions were correctly labeled as patients (recall) and what was the accuracy with regards to correctly predicted patients (precision)?
Mean Square Error	$\frac{1}{N}\sum actual-predicted$	What is the absolute difference between the true scores and the predicted scores?
Root Mean Square Error	$\sqrt{\frac{\sum_{i=1}^{N}{\left({Predicted}_{i}-{Actual}_{i}\right)}^{2}}{N}}$	What is the average difference between the true and the predicted scores (in the same unit of the true scores)?
R2	$1-\frac{RSS}{TSS}$	What fraction of the variance in the data is captured by the model?

## Data Availability

No new data were created or analyzed in this study. Data sharing is not applicable to this article.

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
