# Peer review of "Machine Learning Techniques for Developing Remotely Monitored Central Nervous System Biomarkers Using Wearable Sensors: A Narrative Literature Review"

_sensors, 2023, doi:10.3390/s23115243_

Round 1

Reviewer 1 Report

This review provides a decent overview of machine learning methodology for "beginners", particularly with respect to human health applications. Though this article is in a fairly good state, I have many concerns that should be addressed prior to publication as well as suggestions that may improve the article. Major Comments: - Section 1.1 (Page 1, Line 34): "Patient-generated data from smartphones, wearables, and other remote monitoring devices can potentially complement or substitute clinical visits and enable the development and validation of novel biomarkers." I am quite skeptical that these technologies would substitute (routine) clinical visits. Patient examination/interaction and contextual interpretation of model results will remain/be import components of care. The promise of digital/mHealth technologies is to provide more sensitive and ecologically valid measurements and to increase measurement frequency, thereby "filling in the gaps" and supplementing clinical visits. However, one may potentially expect a reduction of required clinician visits in clinical trials. - The authors should include a subsection about feature standardization methods (e.g., as part of feature engineering). Many training algorithms are sensitive to the scale/distribution of feature values (even in deep models). In particular, it should be noted that PCA (Section 5.3) is sensitive to feature scale. I.e., features with larger scales may naturally have larger variance and thus could "incorrectly" dominate the identified PCs. - Table 1: The authors should mention in this table that "feature engineering" is often considered to be baked into the model architecture and training process in deep learning methods, though significant preprocessing may take place beforehand. - Table 2 / Table 3: "The participants must have a clinical diagnosis that is affected by the CNS." How was this list of diagnoses / search terms for inclusion in the review constructed? - Section 3.1 (Page 7, Line 167): "We found that less than half of the studies included healthy controls (N = 34)." This finding seems a great opportunity to suggest (later in the paper) how critical the inclusion of normative data can be. Even for a study focused on severity estimation, normative data helps to determine if the identified features/patterns are actually related to severity and it is often illuminating to examine how a model performs on controls. It is also ideal if normative data is age-matched, gender-matched, etc... - Section 5.5 (Page 12, Line 322): "As the constructed times series features may not have clinical relevance, the inability to interpret the abstracted features may not be an impediment." I am not quite sure what they authors are trying to say here. On a related note, even PCA-based features derived from expert features may not have clinical relevance. (E.g., they may represent real trends in the data capturing fundamental physical relationships that are not affected by the disease of interest). - Section 5.6: The authors should present a more comprehensive and nuanced overview of over- and under-fitting. These conditions can arise from a variety of sources, including the representational capacity of a model (e.g., number of parameters, linear v.s. nonlinear transformations), overly weak/strong regularization, issues with the training data (e.g., not enough for a given model, a variance/distribution mismatch with the validation/testing data), regularizing effects of data augmentation strategies, etc... - Section 7.6: Though deep models may not be as interpretable as other models, there IS literature describing methods to understand what is going on inside the black box. Saliency maps, for example. - Section 7.9 / Section 10: It should be made clear that in order to properly validate generalized models, a participant's data cannot appear in both the training and testing sets. E.g., if a participant has multiple observations, then cross validation at the observation level as opposed to the participant level is a severe methodological flaw. This issue is a common trap and should be explicitly called out. For an example article related to this effect, see: Gholamiangonabadi, Davoud, Nikita Kiselov, and Katarina Grolinger. "Deep neural networks for human activity recognition with wearable sensors: Leave-one-subject-out cross-validation for model selection." IEEE Access 8 (2020): 133982-133994. - Section 8: An important detail (and potential beginner trap) for hyperparameter search algorithms is the distribution of the search space. E.g., regularization strengths are often exponentially/logarithmically-spaced as opposed to linearly-spaced. - Section 9.1: Binary classification models tend to inherently have a decision threshold hyperparameter (e.g., the model outputs a probability/score and will by default return a positive label if p > 0.5 or score > 0). Metrics such as the ROC curve and precision-recall curve can be quite useful assessments of model performance because they show model performance across all possible decision thresholds. It is also worth noting that many classification metrics assume binary labels, and there are a variety of ways for adapting them to assess model performance on multiclass problems. - Section 9.1 / Section 9.2: It is often good practice to track multiple metrics and make use of model performance visualizations (e.g., confusion matrices for classification or scatter plots / 2D histograms of label vs. estimated values for regression). This practice helps to avoid the "blind spots" of trusting any single metric. - Section 9.2 (Page 21, Line 741): "Furthermore, it's not always suitable to compare R^2 across different datasets as different clinical populations are likely to differ in their feature variance. A disadvantage is that the R^2 will increase with the number of features [181]." The cited paper seems to be the case of looking at R^2 for each feature independently, which is not how one would assess the performance of a composite model. One could instead examine the R^2 between label variance and estimation/output variance (similarly to other metrics, e.g., MSE). - This review reminded me of the following related reviews/articles. The authors may wish to consider these papers and their recommendations when revising this article. References for further reading are likely appropriate in several cases. Halilaj, Eni, et al. "Machine learning in human movement biomechanics: Best practices, common pitfalls, and new opportunities." Journal of Biomechanics 81 (2018): 1-11. Petersen, Eike, et al. "Responsible and regulatory conform machine learning for medicine: A survey of challenges and solutions." IEEE Access 10 (2022): 58375-58418. Yang, Jenny, Andrew AS Soltan, and David A. Clifton. "Machine learning generalizability across healthcare settings: insights from multi-site COVID-19 screening." NPJ Digital Medicine 5.1 (2022): 69. Agarwal, Smisha, et al. "Guidelines for reporting of health interventions using mobile phones: mobile health (mHealth) evidence reporting and assessment (mERA) checklist." BMJ 352 (2016). Artusi, Carlo Alberto, et al. "Implementation of mobile health technologies in clinical trials of movement disorders: underutilized potential." Neurotherapeutics 17 (2020): 1736-1746. Dobkin, Bruce H., and Andrew Dorsch. "The promise of mHealth: daily activity monitoring and outcome assessments by wearable sensors." Neurorehabilitation and Neural Repair 25.9 (2011): 788-798. Suggestions: - Section 1.2 (Page 2, Line 57): "An ML algorithm consists of mathematical equations and instructions to train an ML model, which can then learn complex patterns from available datasets." For clarity, it may be worth disentangling the model and the training algorithm. The model is a set of tunable parameters and an algorithm for generating outputs given inputs and those parameters. ("Outputs" still applies to unsupervised methods in a sense, as they produce an encoding/latent representation.) There is a separate algorithm that learns appropriate values of the model parameters based on the given training data. I think the authors have tried to do this to some extent, but it was still a bit muddy for me. - Section 1.2 (Page 2, Line 65): "Although ML models may not explicitly identify causal relationships between the variables and outcomes, the models can still provide relevant inferential insights." The authors may wish to make this a stronger statement, in that most ML models in use are "correlation engines" and do not, in and of themselves, derive causal insights. - In relation to the above comment, the authors do talk about interpretability, but it may be worth further emphasizing that, even in black box models, it is quite important to ensure that the model is doing something reasonable. For example, a poor model may still achieve "good performance" if the testing data does not penalize predictions based on irrelevant, superficial, or misleading features (e.g., the absence/presence of surgical ink when trying to detect the presence of a tumor in an image). - Table 1: For the model validation example, I would consider including some regression performance metrics. - Figure 3: For the N of this review, it may aid visualization to overlay a swarm plot / strip plot on the box plots. If doing so, it may then be useful to only included the box plots for categories with sufficient (e.g., N > 5) data. - Section 5.5: It may be worth noting that deep learning models can be particularly "data hungry", though this in part depends on model architecture and the associated inductive biases. On another note, recent research (e.g., in the multi-omics space but also relevant to wearables) has shown that deep models can be very useful for jointly modeling different but concurrent/related data modalities. - Section 7: It is worth noting that unsupervised and supervised methods are not mutually exclusive and can be used in tandem. E.g., one could use unsupervised methods to identify a useful latent representation of the input data and then use the latent vectors and inputs to a supervised model. This is what is happening when employing PCA (or other dimensionality reduction techniques). A combination of unsupervised and supervised can also be useful in other scenarios (e.g., mixed loss functions for deep networks, automated outlier rejection). - Section 7.5: Perhaps note that lasso/ridge/elastic-net are all essentially linear regression with different regularization terms in the loss function. - Section 7.7: Transfer learning can also be useful when trying to make more efficient use of rarer data. E.g., one could pretrain a model on readily available normative data to learn the general structure of the data and then fine-tune the model on the population of interest. Another application of this idea is model personalization. - Section 8 (Page 19, Line 638): I would consider adding a few more examples to demonstrate the variety of what may constitute a hyperparameter. E.g., the number of trees in a Random Forest, the number and width of layers in a feed-forward neural network, or the feature selection algorithm used can all be considered as model hyperparameters if not fixed a priori. Typos/Grammar: - The table/figure linking appears to be broken. "Error! Reference source not found." appears multiple times throughout the manuscript. - The manuscript would benefit from another proofreading pass. E.g., in the abstract, "... however further research and study design *standardized* are needed to advance this field." - Section 1.3.2 (Page 4, Line 128): The text is repetitive. "The intervention and device criteria were limited to passive data collected from smartphones and other non-invasive remote monitoring sensors, whereas data collected using active engagement from participants, such as disposable blood tests or small scales, were excluded. Data collection that required active engagement from participants (e.g., disposable blood tests or smart scales) was not considered."

Reviewer 2 Report

1.      The abstract is written in a generic manner. I suggest reorganizing the abstract to provide a clearer structure and improve readability. I recommend including the following elements in the specified order: background of the research, objectives, methodology, results, and conclusion. This reorganization will help readers better understand the flow of your research.

2.      Introduction section has also not been written in an effective manner. There is a lack of transparency concerning the beginning of the research that is associated with Machine learning techniques and its application for developing remotely monitored central nervous system biomarkers. The author (or authors) needs to make the need for the subject very obvious. In this manner, the article's objective will be made obvious, and it will provide a perspective that is more thorough.

3.      The objectives are also not clear. These has to be more specific.

4.      I can’t see any reference to the source of table 1. Is it author’s own?

5.      Kindly look at these sentences: in section 2 and section 2.1.

a.       The inclusion and exclusion criteria were defined by the Population, Intervention, 123 Comparator, Outcomes, Study type (PICOS) framework [13] (Error! Reference source not 124 found.).

b.      First, we conducted an initial limited search of online data- 141 bases for clinical studies that use digital phenotyping and ML. Next, we developed a cus- 142 tom keyword strategy and performed a second in-depth search in PubMed, IEEE Xplore, 143 and CTTI (Error! Reference source not found.).

6.      Methodology is not clear. It would be better if the authors could provide a framework to represent the methodology followed. Currently authors highlighted the literature search process as the methodology.

7.      Section headings of section 5, 6, 7 and 8, 9, 10 are confusing, these are part of results and discussion section what I believe.

8.      Provide a deeper reflection on the research limitations and suggestions for future studies(See: https://doi.org/10.3390/ijerph20043222;  https://doi.org/10.3390/su141811698; https://doi.org/10.3390/su14031904

minor English check required

Round 2

Reviewer 1 Report

The authors have addressed my concerns.